# Research on Coil Impedance of Self-Inductive Displacement Sensor Considering Core Eddy Current

**DOI:** 10.3390/s21186292

**Published:** 2021-09-20

**Authors:** Zongqiang Ren, Hongwei Li, Wentao Yu

**Affiliations:** School of Electrical Engineering, Shandong University, Jinan 250061, China; 202034695@mail.sdu.edu.cn (Z.R.); wintop@sdu.edu.cn (W.Y.)

**Keywords:** self-inductive displacement sensor, core eddy current, complex permeability, improved theoretical model, sensitivity

## Abstract

The inductive displacement sensor is widely used in active magnetic bearing (AMB) systems to detect rotor displacement in real time, and the performance of the sensor directly affects the performance of AMB. At present, most theoretical studies on the working principle of inductive displacement sensor are based on a traditional mathematical model, ignoring the influence of the core magnetic resistance and core eddy current, which will lead to a certain error between the theoretical analysis of the sensor output characteristics and the actual situation. In this regard, based on the theory of electromagnetic field and circuit, an improved theoretical model of the inductive sensor was established in this paper by introducing the complex permeability, by which the influence of core eddy current on magnetic field can be taken into account. In order to verify the improved model, an eight-pole radial self-inductive displacement sensor with an air gap of 1 mm was designed. Then the electromagnetic field of the designed sensor was simulated by a finite element software and the GW LCR-6100 measuring instrument was used to measure the changes of the inductance and resistance of the designed sensor core coils with the rotor displacement at 20–100 kHz. The results demonstrated that there is a good linear relationship between the impedance change of the sensor coils and the rotor displacement within the measurement range of −0.4 ~ +0.4 mm. At the same time, compared with the traditional model, the sensitivity of the improved theoretical model is closer to the results from FEM and experiment, and the accuracy of the sensitivity of the improved theoretical model can be approximately doubled, despite there are certain differences with the experimental situation. Therefore, the improved theoretical model considering complex permeability is of great significance for studying the influence of core eddy current on the coil impedance of sensor.

## 1. Introduction

Active magnetic bearing (AMB) has the advantages of no contact, no friction, no wear, no need for lubrication, high rotor speed, low power consumption, simple maintenance, and long life. It has a very broad application prospect in the fields of aerospace, mechanical engineering, and life sciences [1,2,3,4,5].

The displacement sensor is an important part of the AMB closed-loop control system. It controls the stable suspension of the rotor by detecting the displacement of the magnetic suspension rotor in real time, and its performance directly affects the performance of the AMB. At present, the displacement sensors widely used in AMB include eddy current displacement sensor, laser displacement sensor, and inductive displacement sensor. The eddy current displacement sensor has the advantages of simple structure, low cost, high sensitivity, and a fast response speed [6], but it is very sensitive to the material and size of the measured object, and is susceptible to interference from external magnetic field. The laser displacement sensor has the advantages of high measurement accuracy, small size, and flexible structure design [7], but the cost is relatively high. The inductive displacement sensor has the advantages of strong anti-interference, high sensitivity, large linear measurement range, and long signal transmission distance. Moreover, the mechanical structure of inductive displacement sensor is close to the structure of AMB. Therefore, the inductive displacement sensor is more and more widely used in an AMB system [8,9,10,11,12].

At present, electromagnetic field modeling and analysis has been an important research content in the field of magnetic bearing [13,14,15]. The establishment of an accurate theoretical model is the basis for the design, structural optimization, and excitation frequency selection of the inductive displacement sensor. It is also of great significance to the theoretical research on the output characteristics of the sensor.

In the process of theoretical research on the inductive displacement sensor, it is generally considered that the reluctance of the air gap is much greater than the reluctance of the stator and rotor cores, and the influence of the core reluctance is often ignored, so that the relationship between the output voltage and the rotor displacement can be roughly obtained in theory. In practical applications, in order to improve the response speed, the excitation frequency of the inductive displacement sensor usually needs to reach tens of kHz. At this time, the influence of the core eddy current cannot be ignored. However, the traditional theoretical model cannot intuitively reflect the influence of core eddy current, so it is necessary to improve the theoretical model.

The research on core eddy current has always attracted the attention of scholars at home and abroad, but current research mainly has two ideas. One is based on the physical field, by analyzing the influence of eddy current on the magnetic field to study the eddy current loss [16,17,18,19,20]. The other is based on the mathematical description of material behavior, attributing the effect of eddy current to changes in material properties, and studying the effect of eddy current by analyzing the changes in the materials’ permeability and conductivity [21,22,23,24,25,26].

At present, the research on core eddy current loss is relatively mature, and it is basically possible to accurately analyze the eddy current loss under high frequency. In 2002, Sun et al. studied the eddy current loss in a magnetic bearing laminated rotor, determined the magnetic field according to the magnetic flux distribution on the surface of the magnetic pole, and established a magnetic circuit model considering the eddy current [16]. The rotational loss caused by the induced eddy current in the rotor lamination is approximately analyzed by this method, which is basically consistent with the experimental loss. In 2021, Tong et al. studied the eddy current loss of high-frequency axial flux permanent magnet motors, coupled the accurate subdomain method with the resistance network model, and established a three-dimensional permanent magnet eddy current loss analytical model [20]. This method studies the reaction of eddy current to the magnetic field based on the three-dimensional eddy current distribution density, and then estimates the eddy current loss. Compared with the two-dimensional eddy current loss analytical model, it has higher calculation accuracy. The above-mentioned documents are all research from the perspective of the influence of eddy current on the magnetic field. The theoretical method can help the loss analysis of the inductive displacement sensor, but it can only do a quantitative study on the output characteristics of the sensor, and cannot qualitatively analyze the influence of eddy current on the coil inductance change of the inductive displacement sensor.

Therefore, in order to qualitatively study the influence of the eddy current on the coil inductance of the inductive displacement sensor, the influence of the eddy current on the magnetic field can be transformed into the influence on the magnetic permeability of the core material, and the influence of the eddy current on the coil inductance can be reflected by studying the change of the magnetic permeability of the sensor core material.

Analyzing the magnetic field by attributing the effect of eddy current to changes in material properties is a new method. Many scholars have studied this method and put forward the concept of complex permeability. In 2010, Mitchell et al. used core permeability to study the influence of eddy current on the frequency response of transformers. According to Maxwell’s equation, they derived a theoretical model of complex permeability that considered the influence of eddy current [22]. Through experiments, it was found that at high frequencies, the transformer model introduced with complex permeability has higher modeling accuracy, which is helpful for the frequency response analysis of the transformer. In 2015, Hamzehbahmani et al. analyzed the eddy current loss in electrical steel sheet, established the eddy current power loss model of magnetic laminations, and studied the influence of the magnetic permeability of the material on the magnetic properties of electrical steel [24]. It was found that, in order to analyze the complex magnetization phenomenon in the magnetic core, the relative permeability of the material is regarded as the complex permeability, which can more effectively analyze the influence of eddy current under the high-frequency magnetic field and greatly improve the analysis accuracy of the model. In 2017, Yang et al. studied the impedance characteristics of non-contact magnetic position sensors using a finite element modeling method that considered the complex permeability [25]. It was found that at high frequencies, this method can effectively reflect the influence of eddy current, and has a certain guiding effect on research.

Therefore, based on the existing research, this paper established the coil impedance model of self-inductive displacement sensor that considered the complex permeability. At the same time, a self-inductive displacement sensor was designed in the laboratory, and the improved theoretical model was experimentally analyzed. The improved theoretical model was used to study the change of the coil impedance relationship with frequency and rotor displacement, and compared it with the traditional theoretical model to verify that it could better reflect the relationship between sensor sensitivity and eddy current.

## 2. Theoretical Analysis

### 2.1. Theoretical Model of Self-Inductive Displacement Sensor

The inductive displacement sensor is divided into two types: self-inductive and mutual inductive. This paper mainly studies the self-inductive displacement sensor. The basic structure of the self-inductive displacement sensor is shown in Figure 1. The mechanical part is mainly composed of three parts: the coils, the stator core, and a movable rotor core.

Since the air gap between the stator and the rotor is smaller than the size of the stator poles, it can be approximated that the magnetic field in the air gap is uniform. In the case of ignoring magnetic flux leakage, when the rotor is in the middle position, the total magnetic resistance of the unilateral magnetic circuit is:(1)$$R_{m}=\frac{l_{1}}{\mu_{stator}\mu_{0}A_{1}}+\frac{l_{2}}{\mu_{rotor}\mu_{0}A_{2}}+\frac{2\delta_{0}}{\mu_{0}A_{3}}$$
where *μ*_0_ is the permeability of vacuum. *μ_stator_* and *μ_rotor_* are the relative permeability of stator core and rotor core respectively. *l*_1_ and *l*_2_ are the average length of magnetic circuit in single stator and rotor respectively. *δ*_0_ is the length of one side air gap. *A*_1_, *A*_2_, and *A*_3_ are the cross-sectional areas of stator core, rotor core, and air gap respectively.

In general, it is considered that the magnetic circuit cross-sectional areas of stator and rotor cores are equal, and the stator and rotor cores are of the same material. The cross-sectional area of the air gap is *b* times that of the magnetic circuit of stator and rotor cores. So, take *A*_3_ = *bA*_1_ = *bA*_1_ = *bA*_0_, *μ*^*^ = *μ_stator_* = *μ_rotor_*, *l* = *l*_1_ + *l*_2_.

According to Ohm’s law of the magnetic circuit, the self-inductance expression of the core coil is:(2)$$L=\frac{\Psi }{I}=\frac{N\Phi }{I}=\frac{N^{2}}{R_{m}}$$
where *ψ* is the magnetic link in the magnetic circuit. *Φ* is the magnetic flux in the magnetic circuit. *I* is the coil current. *N* is the number of coil turns.

Therefore, the inductance of the core coil can be expressed as:(3)$$L=\frac{N^{2}}{\frac{l}{\mu_{0}\mu^{*}A_{0}}+\frac{2\delta_{0}}{\mu_{0}\cdot bA_{0}}}=\frac{L_{0}}{\frac{1}{b}+\frac{l}{2\delta_{0}}\cdot \frac{1}{\mu^{*}}}$$
where *L*_0_ is the coil inductance when the cross-sectional area is *A*_0_, the rotor is in the middle position and the core reluctance is not considered, namely:(4)$$L_{0}=\frac{\mu_{0}A_{0}N^{2}}{2\delta_{0}}$$

Take *k*_1_ = 1/*b*, *k*_2_ = *l*/2*δ*_0_. Since the resistance of the copper wire of the coil is small and can be ignored, the impedance of the core coil can be simplified as:(5)$$Z=R_{0}+j\omega L\approx j\omega L=j\omega L_{0}\cdot \frac{1}{k_{1}+k_{2}\cdot \frac{1}{\mu^{*}}}$$
where *R*_0_ is the resistance of the copper wire of the coil, and *ω* is the excitation frequency of the external power supply.

### 2.2. The Influence of Core Eddy Current on Core Permeability

In order to intuitively reflect the influence of core eddy current, the influence of the eddy current on the magnetic field can be transformed into the influence on the magnetic permeability of the iron core for research.

Since the thickness of the laminated silicon steel sheet is much smaller than its length and width, it can be approximated that the eddy current *J_x_* flows only near the surface of the conductor.

Therefore, the eddy current in the sensor core lamination can be studied in the way shown in Figure 2.

Under the condition of a sinusoidal alternating magnetic field, according to Faraday’s law of electromagnetic induction, ignoring the influence of displacement current and interlayer current, the diffusion equation can be written as:(6)$$\frac{\partial^{2}H_{z}}{\partial y^{2}}=j\omega \sigma \mu_{0}\mu_{r}H_{z}=k^{2}H_{z}$$
where *k* is the propagation constant. *σ* is the conductivity of the core. *μ_r_* is the relative permeability of the core material.

For the core laminate of the sensor, the magnetic field intensity does not attenuate basically on the surface of the laminate. Based on the electromagnetic field theory [22], the average magnetic field intensity of the core lamination is:(7)$${\bar{H}}_{z}=\frac{k\Phi \cosh(k\cdot \frac{d}{2})}{2\mu_{0}\mu_{r}L_{s}\sinh(k\cdot \frac{d}{2})}$$

In the *z* direction, the average magnetic flux density through the cross-sectional area of the laminate is:(8)$$\bar{B}=\frac{\Phi }{L_{s}d}$$

Combined with the Formulas (6)–(8), the relative complex permeability considering the influence of core eddy current can be obtained:(9)$$\mu^{*}=\frac{\bar{B}}{\mu_{0}{\bar{H}}_{z}}=\mu_{r}\frac{\tanh\left(k\cdot \frac{d}{2}\right)}{k\cdot \frac{d}{2}}=\mu_{1}-j\mu_{2}$$
where *μ*_1_ is the real part of the relative complex permeability, which reflects the magnetic energy storage capacity of the core material. *μ*_2_ is the imaginary part of the relative complex permeability, which reflects the magnetic energy loss ability of the core material.

### 2.3. The Influence of Core Eddy Current on Coil Impedance

In order to reflect the influence of core eddy current through the coil inductance, substitute *μ*^*^ = *μ*_1_ − *jμ*_2_ into the formula (3), and the coil inductance considering the complex permeability can be obtained as:(10)$$L=\frac{L_{0}}{k_{1}+k_{2}\cdot \frac{1}{\mu_{1}-j\mu_{2}}}=L_{0}\cdot \left[\frac{k_{1}\left(\mu_{1}^{2}+\mu_{2}^{2}\right)+k_{2}\mu_{1}}{{\left(k_{1}\mu_{1}+k_{2}\right)}^{2}+{\left(k_{1}\mu_{2}\right)}^{2}}-j\frac{k_{2}\mu_{2}}{{\left(k_{1}\mu_{1}+k_{2}\right)}^{2}+{\left(k_{1}\mu_{2}\right)}^{2}}\right]$$

Therefore, the impedance of the core coil is:(11)$$Z=j\omega L=\omega L_{0}\cdot \frac{k_{2}\mu_{2}}{{\left(k_{1}\mu_{1}+k_{2}\right)}^{2}+{\left(k_{1}\mu_{2}\right)}^{2}}+j\omega L_{0}\cdot \frac{k_{1}\left(\mu_{1}^{2}+\mu_{2}^{2}\right)+k_{2}\mu_{1}}{{\left(k_{1}\mu_{1}+k_{2}\right)}^{2}+{\left(k_{1}\mu_{2}\right)}^{2}}=R_{eq}+j\omega L_{eq}$$
where *R_eq_* is the equivalent AC resistance of the coil. *L_eq_* is the equivalent AC inductance of the coil.

The ratio relationship between the equivalent inductance and the equivalent resistance of the core coil is:(12)$$\frac{\omega L_{eq}}{R_{eq}}=\frac{k_{1}}{k_{2}}\cdot \frac{\mu_{1}^{2}+\mu_{2}^{2}}{\mu_{2}}+\frac{\mu_{1}}{\mu_{2}}$$

*k*_1_ and *k*_2_ are mainly determined by the structure of the sensor and the air gap, which are basically fixed values. Therefore, the ratio of inductance to resistance is affected by the change of complex permeability.

### 2.4. Radial Displacement of the Rotor

When the rotor core is radially displaced, the air gap between the stator and rotor cores will change. Taking the upward movement of the rotor as an example, the air gap above the rotor becomes *δ*_1_ = *δ*_0_ − Δ*δ*, and the air gap below the rotor becomes *δ*_2_ = *δ*_0_ + Δ*δ*. At this time, the total reluctance in the magnetic circuit changes, causing the inductance of the core coil to change.

In the traditional theoretical model, it is generally considered that the magnetic resistance of the stator and rotor cores is much greater than the air gap magnetic resistance, so that the core magnetic resistance is ignored, that is, the formula (3) is approximated as:(13)$$L\approx \frac{N^{2}}{\frac{2\delta_{0}}{\mu_{0}\cdot bA_{0}}}=\frac{L_{0}}{k_{1}}$$

Then, the inductances of the upper and lower core coils become:
(14)$$\left\{ \begin{array}{l} L_{1}=\frac{L_{0}}{k_{1}\left(1-\frac{\Delta \delta }{\delta_{0}}\right)}\\ L_{2}=\frac{L_{0}}{k_{1}\left(1+\frac{\Delta \delta }{\delta_{0}}\right)} \end{array}\right.$$

The equivalent circuit of the differential inductive sensor is shown in Figure 3.

Therefore, through the traditional model, the relationship between the output voltage amplitude and the rotor displacement can be roughly obtained as:(15)$$\Delta U=U_{1}-U_{2}=\frac{Z_{1}-Z_{2}}{Z_{1}+Z_{2}}U_{0}=\frac{L_{1}-L_{2}}{L_{1}+L_{2}}U_{0}=\frac{\Delta \delta }{\delta_{0}}U_{0}$$
where
(16)$$\left\{ \begin{array}{l} Z_{1}=R_{0}+{j\omega L}_{1}\\ Z_{2}=R_{0}+{j\omega L}_{2} \end{array}\right.$$

However, for the theoretical model considering the core eddy current, the inductances of the upper and lower core coils become:(17)$$\left\{ \begin{array}{l} L_{1}=\frac{L_{0}}{k_{1}\left(1-\frac{\Delta \delta }{\delta_{0}}\right)+k_{2}\cdot \frac{1}{{\mu^{\prime }}^{*}}}\\ L_{2}=\frac{L_{0}}{k_{1}\left(1+\frac{\Delta \delta }{\delta_{0}}\right)+k_{2}\cdot \frac{1}{{\mu^{\prime \prime }}^{*}}} \end{array}\right.$$
where *μ*′* is the relative complex permeability of the upper core, *μ*″* is the relative complex permeability of the lower core.

At this time, the amplitude of the output voltage is:(18)$$\Delta U=\left|\frac{Z_{1}-Z_{2}}{Z_{1}+Z_{2}}\right|U_{0}=\left|\frac{L_{1}-L_{2}}{L_{1}+L_{2}}\right|U_{0}=mU_{0}$$

According to the Formulas (17) and (18), *m* is related to the change of air gap and complex permeability.

It can be seen that in the traditional theoretical model, the output voltage is only related to the input voltage and the air gap, and has nothing to do with the excitation frequency.

However, for the self-inductive displacement sensor, the excitation frequency can reach tens of kHz, which will produce a non-negligible core eddy current effect. Therefore, it is of great significance to improve the traditional model to study the relationship between the output voltage and the input voltage at different frequencies. The improved theoretical model can reflect the influence of the core eddy current on the output voltage through the complex permeability.

## 3. Establishment of the Improved Theoretical Model

### 3.1. Estimation of Constant Parameters in the Improved Theoretical Model

The laboratory has designed an 8-pole self-inductive displacement sensor with air gap of 1 mm. Its structural parameters are shown in Table 1, and the structure is shown in Figure 4.

The stator and rotor cores of the designed sensor are made of WG35WW250 silicon steel sheets, and the stator core is made of 10 silicon steel sheets of the same structure and size, with a total thickness of 4mm. According to the designed sensor structure, it is estimated that the average path of the stator and rotor magnetic circuit is roughly 25 times the total length of the magnetic circuit in the air gap, that is, *k*_2_ = 25.

At the same time, there will be a certain edge effect between the stator and the rotor. Taking the air gap above the rotor as an example, the magnetic flux distribution in the air gap is shown in Figure 5.

According to the diffraction effect, the average cross-sectional area of the air gap magnetic circuit is approximately 1.1 to 1.3 times the cross-sectional area of the stator and rotor magnetic circuit. Therefore, *k*_1_ = 0.85 can be taken.

### 3.2. Estimation of the Relationship between the Real and Imaginary Parts of the Relative Complex Permeability

For the self-inductive displacement sensor designed in the laboratory, a 2 V sinusoidal voltage source will be given for excitation in the subsequent experiment. Its magnetic field intensity is small, and the magnetic flux density *B* does not exceed 0.2 T. According to the *B-H* relationship of the WG35WW250 silicon steel sheet used in the sensor core, as shown in Figure 6, the relative permeability *μ_r_* of the core material is estimated to not exceed 3000. The conductivity of the silicon steel sheet is about 1.85 × 10^6^ S/m. It is assumed that *μ_r_* is a constant of 3000. Further, according to the structure of the sensor core and combined with the formula (9), the variation trend of *μ*_1_ and *μ*_2_ with the excitation frequency, which considers the core eddy current, is obtained by MATLAB as shown in Figure 7.

The excitation frequency of the sensor is usually between 20–100 kHz. It can be seen from Figure 7 that the relative complex permeability in this case is small and the coincidence of the real and imaginary parts is high. Taking *μ* = *μ*_2_ = *aμ*_1_, the relationship between *a* and frequency can be obtained as shown in Figure 8.

In practice, the relative permeability *μ_r_* of the sensor core material will be affected by the change of magnetic field, so *μ_r_* will fluctuate around 3000. Therefore, in order to further study the influence of different *μ_r_* on the correlation between *a* and frequency, based on the above, choose *μ_r_* between 2000 and 5000 for research, and the relationship between *a* and frequency is shown in Figure 9.

According to Figure 9, for the designed sensor, in the theoretical model considering core eddy current, it can be approximated that the real and imaginary parts of the sensor core’s complex permeability are the same, that is, *a* = 1.

### 3.3. Estimation of Core Coil Impedance

According to the estimated constant parameters and the relationship between the real and imaginary parts of the relative complex permeability, when the rotor is in the middle position, the impedance of the core coil of the sensor can be approximately as follows:(19)$$\begin{array}{l} Z=\omega L_{0}\cdot \frac{25\mu }{{\left(0.85\mu +25\right)}^{2}+{\left(0.85\mu \right)}^{2}}+j\omega L_{0}\cdot \frac{1.7\mu^{2}+25\mu }{{\left(0.85\mu +25\right)}^{2}+{\left(0.85\mu \right)}^{2}}\\ =\omega L_{0}\cdot \frac{25\mu }{1.445\mu^{2}+42.5\mu +625}+j\omega L_{0}\cdot \frac{1.7\mu^{2}+25\mu }{1.445\mu^{2}+42.5\mu +625} \end{array}$$

When the rotor moves up, the magnetic field in the upper core will increase, and the magnetic field in the lower core will decrease. According to the *B-H* relationship of the silicon steel sheet, the relative permeability of the upper and lower cores will change to a certain extent. According to the formula (9), when the relative permeability changes, it will also affect the relative complex permeability, thus affecting *μ*. Therefore, it can be said that the *μ* of the upper core becomes *μ*′, and the *μ* of the lower core becomes *μ*″. At this time, the upper and lower coil impedances respectively become:(20)$$\left\{ \begin{array}{l} Z_{1}=\omega L_{0}\cdot \frac{25\mu^{\prime }}{1.445{\left(1-\frac{\Delta \delta }{\delta_{0}}\right)}^{2}{\mu^{\prime }}^{2}+42.5\left(1-\frac{\Delta \delta }{\delta_{0}}\right)\mu^{\prime }+625}+j\omega L_{0}\cdot \frac{1.7\left(1-\frac{\Delta \delta }{\delta_{0}}\right){\mu^{\prime }}^{2}+25\mu^{\prime }}{1.445{\left(1-\frac{\Delta \delta }{\delta_{0}}\right)}^{2}{\mu^{\prime }}^{2}+42.5\left(1-\frac{\Delta \delta }{\delta_{0}}\right)\mu^{\prime }+625}\\ Z_{2}=\omega L_{0}\cdot \frac{25\mu^{\prime \prime }}{1.445{\left(1+\frac{\Delta \delta }{\delta_{0}}\right)}^{2}{\mu^{\prime \prime }}^{2}+42.5\left(1+\frac{\Delta \delta }{\delta_{0}}\right)\mu^{\prime \prime }+625}+j\omega L_{0}\cdot \frac{1.7\left(1+\frac{\Delta \delta }{\delta_{0}}\right){\mu^{\prime \prime }}^{2}+25\mu^{\prime \prime }}{1.445{\left(1+\frac{\Delta \delta }{\delta_{0}}\right)}^{2}{\mu^{\prime \prime }}^{2}+42.5\left(1+\frac{\Delta \delta }{\delta_{0}}\right)\mu^{\prime \prime }+625} \end{array}\right.$$

At this time, for the upper and lower coils, the ratios of the equivalent inductance to the equivalent resistance are:(21)$$\left\{ \begin{array}{l} \frac{\omega L_{eq1}}{R_{eq1}}=0.068\left(1-\frac{\Delta \delta }{\delta_{0}}\right)\mu^{\prime }+1\\ \frac{\omega L_{eq2}}{R_{eq2}}=0.068\left(1+\frac{\Delta \delta }{\delta_{0}}\right)\mu^{\prime \prime }+1 \end{array}\right.$$

According to the formula (21), the relationship between inductance and resistance of the core coil is mainly affected by rotor displacement and complex permeability.

## 4. Comparison and Analysis of Theory, Simulation and Experiment

### 4.1. Finite Element Simulation

COMSOL Multiphysics finite element simulation software has the advantages of simple operation and fast calculation speed, and has been widely used by scholars from all over the world [27,28]. Among them, the electromagnetic field simulation is one of the important parts of the software [29].

According to the designed self-inductive displacement sensor, the two-dimensional electromagnetic field finite element model of the sensor as shown in Figure 10 was established in the COMSOL finite element simulation software. According to the actual materials of the sensor, the material parameters such as copper, silicon steel sheet in the COMSOL material library were called to set the material properties of the sensor model, and then the mesh was divided.

In order to compare with the subsequent experiment, the excitation of the sensor was set as a 2 V sinusoidal voltage source, the rotor moved forward along the *X*-axis, the step size was 0.02 mm, and the range was ±0.9 mm. The frequency domain field was used for simulation, and the excitation frequency range was set to 20–100 kHz.

### 4.2. Static Measurement Experiment of Coil Inductance and Resistance

For the designed self-inductive displacement sensor with air gap of 1 mm, the static displacement measurement experiment was carried out with the GW LCR-6100 measuring instrument with the sinusoidal excitation voltage source amplitude of 2 V. The frequency range of the voltage source excitation is 20–100 kHz. The experimental device is shown in Figure 11. During the experiment, the measured values of coil inductance and resistance were recorded every 0.05 mm of rotor movement within the measurement range of ±0.9 mm. The forward and reverse strokes were repeated for six times to calculate the average value.

In practical applications, the output voltage signal needs to be filtered and amplitude modulated to achieve the required sensitivity. Due to different applications, the amplified output voltage range and sensitivity vary greatly. Therefore, this article does not analyze the amplified output voltage. It only estimates the amplitude relationship between the output voltage and the input voltage in the improved model based on the measured coil inductance and resistance. The flow chart is shown in Figure 12.

### 4.3. Comparison and Error Analysis of Theoretical, Simulation and Experimental Results

#### 4.3.1. Comparison of Results

Through simulation, it is found that the distribution of the magnetic flux density modulus obtained by the designed sensor under the frequency domain field is basically the same at different frequencies. Therefore, take the excitation frequency of 50 kHz as an example for analysis, as shown in Figure 13. When the rotor is in the middle position, the magnetic field distribution of the left and right cores of the sensor is relatively uniform. As the rotor moves to the right, the magnetic field of the right core gradually increases and the magnetic field of the left core gradually decreases. The average magnetic flux density of the sensor core varies between 10–40 mT, and there are obvious edge effects at the magnetic poles.

Through experimental research, it is found that within the measurement range of ±0.4 mm, there is a good linear relationship between *m* and the rotor displacement. Taking the excitation frequency of 50 kHz as an example, the relationship between the amplitude of *m* and the rotor displacement under the theoretical, simulation, and experimental conditions is shown in Figure 14.

It can be seen from Figure 14 that as the rotor moves, the *m* of the theoretical model considering the core eddy current is more accurate than that of the traditional theoretical model, and its change is closer to the simulation situation, but there is about a three-fold difference from the experimental results.

According to the definition of sensitivity:(22)$$K=\frac{\Delta U}{\Delta \delta }=\frac{m}{\Delta \delta }U_{0}$$

In the measurement range of ±0.4 mm, the theoretical sensitivity is 2 V/mm when the core reluctance is not considered. For the experiment, simulation and the theoretical model considering core eddy current, the sensitivity and nonlinear error at different frequencies are shown in Table 2, Table 3 and Table 4.

It can be seen from Table 2 that there is a good linear relationship between the output voltage and the rotor displacement within the measurement range of ±0.4 mm for the designed sensor. It can be seen from Table 3 and Table 4 that when the complex permeability considering core eddy current is introduced into the theoretical model of the sensor, there is still a good linear relationship between the output voltage and the rotor displacement within the measurement range of ±0.4 mm, and the sensitivity simulation results at different frequencies are closer to the theoretical results.

At the same time, the variation of experimental sensitivity, simulation sensitivity, and theoretical sensitivity with frequency can be obtained as shown in Figure 15.

It can be seen from Figure 15 that when the theoretical model considering core eddy current is adopted, the variation of sensitivity will be related to the excitation frequency. Comparing the improved theoretical model results with the simulation results, it can be seen that when the excitation frequency is lower than 60 kHz, the sensitivity of the two changes with frequency are relatively close, and both will decrease as the excitation frequency increases. It is showed that the higher the excitation frequency, the greater the eddy current, and the eddy current will weaken the sensitivity of the sensor. However, in the experiment, the sensitivity of the designed sensor is different from the theoretical and simulation results.

Taking the ratio of theoretical sensitivity and experimental sensitivity as *M*, the variation trend of *M* with frequency under the two theoretical models is shown in Figure 16.

It can be seen from Figure 16 that the variation trend of *M* with frequency under the two theoretical models is basically the same, and the sensitivity of the theoretical model considering core eddy current is closer to the actual situation than that of the theoretical model ignoring the core reluctance, and the accuracy of the sensitivity of the improved theoretical model can be approximately doubled.

#### 4.3.2. Error Analysis

According to Figure 14 and Figure 15, it can be found that there are obvious differences between the theoretical and simulation results and the experimental results of the designed self-inductive displacement sensor.

The reason for the above differences is that only the effect of eddy current on core permeability is considered in the process of establishing the theoretical model. The relative permeability *μ_r_* in formula (9) is a real number estimated based on the *B-H* curve of silicon steel. However, in practice, the hysteresis effect will also affect the relative permeability *μ_r_* [30,31]. Therefore, when deriving the relationship between the real and imaginary parts of the core relative complex permeability, the selection of *a* = 1 is estimated on the premise of considering only the eddy current. According to the comparison of theoretical and simulation results, there will be certain errors when choosing the theoretical model with *a* = 1. At the same time, the sensor structure designed in the laboratory is miniaturized, with a magnetic pole width of only 8 mm and a thickness of 4 mm. The balanced air gap of 1 mm is large for the sensor, so the magnetic flux leakage will be serious. At the same time, it can be seen from Figure 7 that when the excitation frequency is above 20 kHz, the relative complex permeability of the silicon steel sheet of the sensor will be greatly attenuated in the case of considering the eddy current. In practice, the permeability of silicon steel sheet will be weakened by the hysteresis effect, and its magnetic property will be more seriously attenuated. Therefore, when the magnetic flux passes through the air gap, it will aggravate the occurrence of magnetic flux leakage. The above-mentioned influences lead to errors between theoretical, simulation, and experimental results.

## 5. Discussion

Through the comparison of theory with simulation and experiment, it is found that the variation of sensitivity with frequency when only considering core eddy current is different from the experimental situation. This is because the excitation frequency of the sensor is between 20–100 kHz, and the actual effect of core hysteresis will also be significant. Moreover, the structure of the designed sensor is miniaturized, and the balanced air gap is relatively large, resulting in serious magnetic flux leakage.

In the future, on the basis of the theoretical model considering core eddy current, the influence of core hysteresis and magnetic flux leakage on the coil impedance of self-inductive displacement sensor will be further studied through the complex permeability, and the theoretical model will be further improved.

## 6. Conclusions

In order to solve the problem that the influence of eddy current on coil impedance cannot be considered in the traditional theoretical model of self-inductive displacement sensor, an improved theoretical model considering core eddy current is proposed, and the following main conclusions are drawn through simulation and static experimental research on the sensor:

(1) The influence of eddy current on the magnetic field can be transformed into the influence on the permeability of the core, so the complex permeability of the core material can be used to study the influence of eddy current on the coil impedance.

(2) The improved theoretical model considering core eddy current can better reflect the relationship between the ratio of the sensor output voltage to the input voltage and the excitation frequency.

(3) Compared with the theoretical model ignoring the core reluctance, the accuracy of the sensitivity of the improved theoretical model can be approximately doubled.

Based on the complex permeability, the improved theoretical model can more intuitively reflect the influence of core eddy current on the coil impedance of the sensor, which is of great significance for the research on the output characteristics of the sensor.

## Figures and Tables

**Figure 1 sensors-21-06292-f001:**
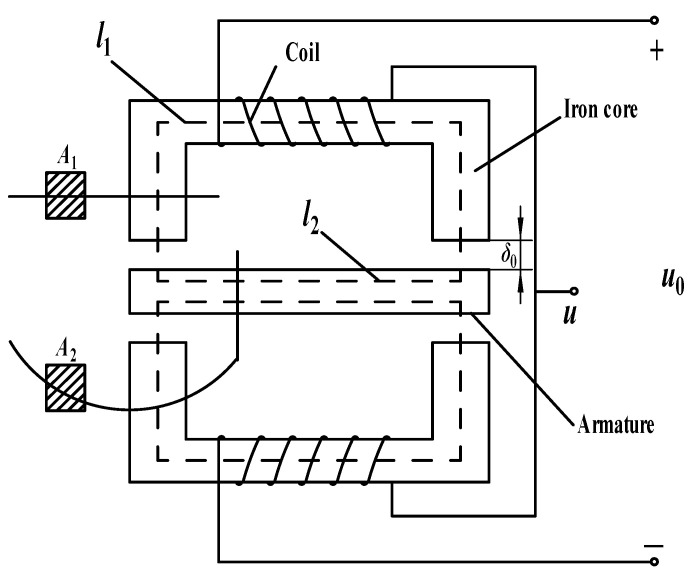
Schematic diagram of the differential inductive sensor structure.

**Figure 2 sensors-21-06292-f002:**
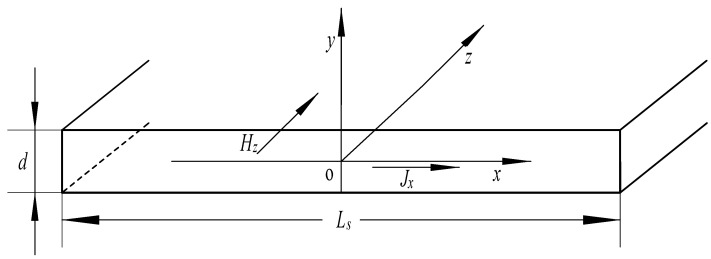
Eddy current model diagram in core lamination. In the figure: *L_s_* is the width of the core lamination, and *d* is the thickness of the core lamination.

**Figure 3 sensors-21-06292-f003:**
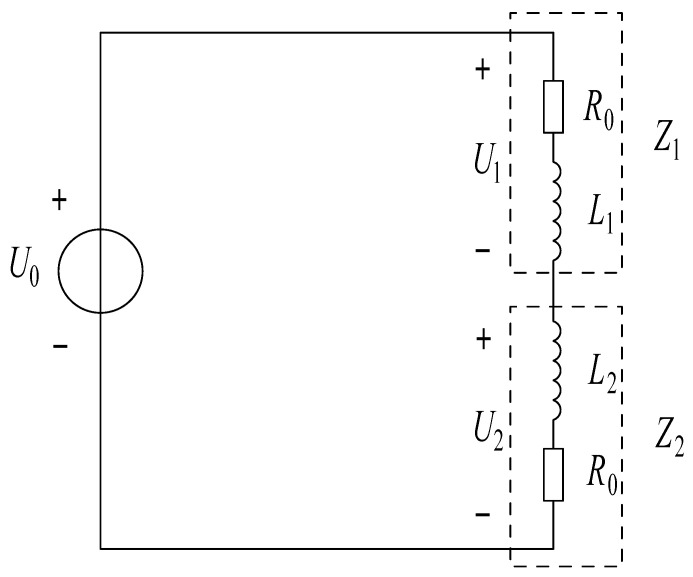
Equivalent circuit of the differential inductive sensor.

**Figure 4 sensors-21-06292-f004:**
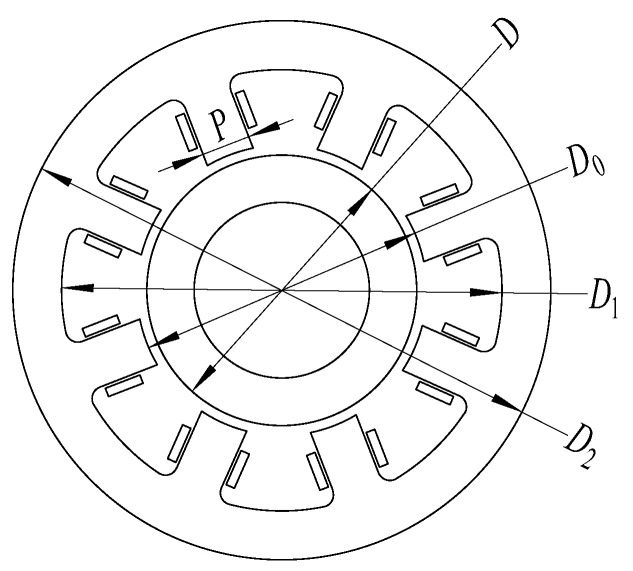
Structure diagram of self-inductive displacement sensor. In the figure: *D*-rotor diameter; *D*_0_-inner diameter of stator; *D*_1_-inner diameter of stator yoke; *D*_2_-outer diameter of stator; *A*_0_-magnetic pole area; *P*-width of pole shoe; *N*_1_-the number of coil turns; *d*_0_-diameter of coil wire.

**Figure 5 sensors-21-06292-f005:**
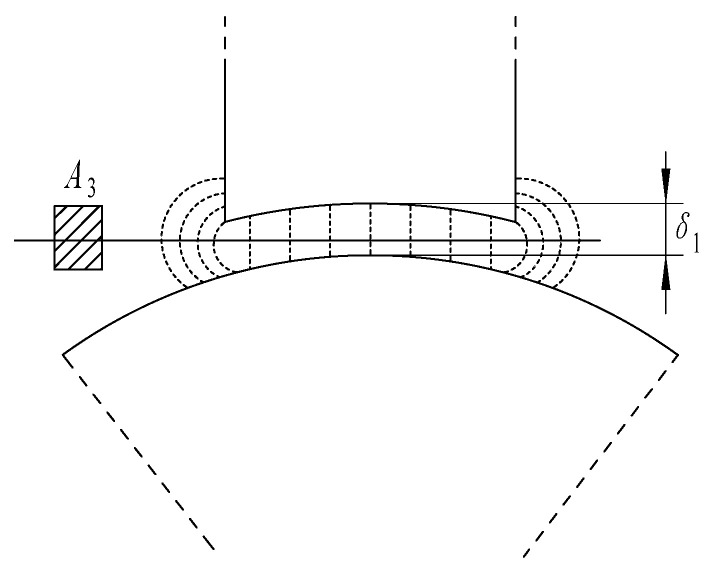
Magnetic flux distribution in the air gap.

**Figure 6 sensors-21-06292-f006:**
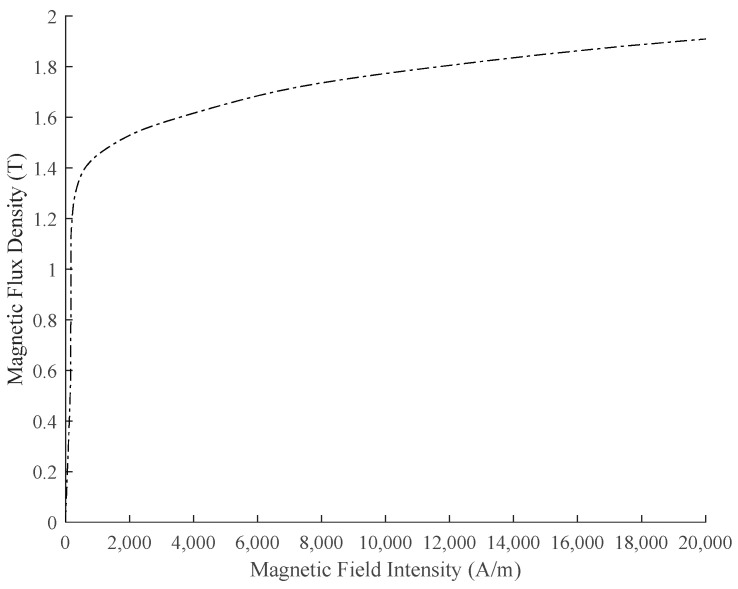
*B**-**H* curve of silicon steel.

**Figure 7 sensors-21-06292-f007:**
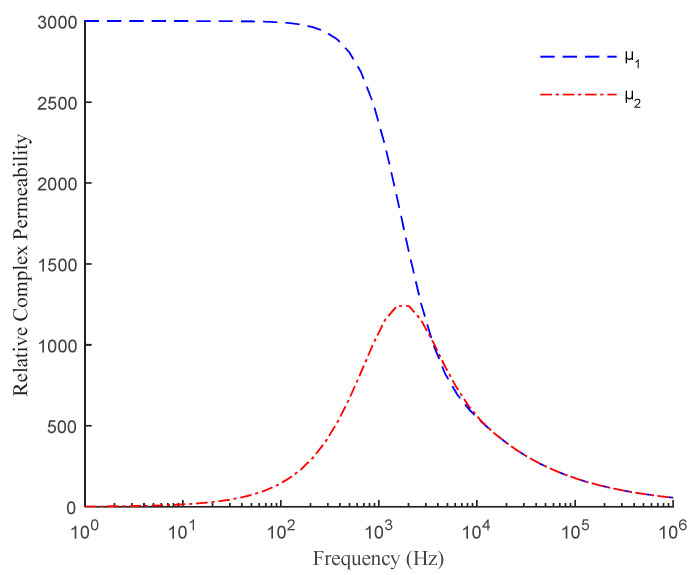
Variation of relative complex permeability with frequency.

**Figure 8 sensors-21-06292-f008:**
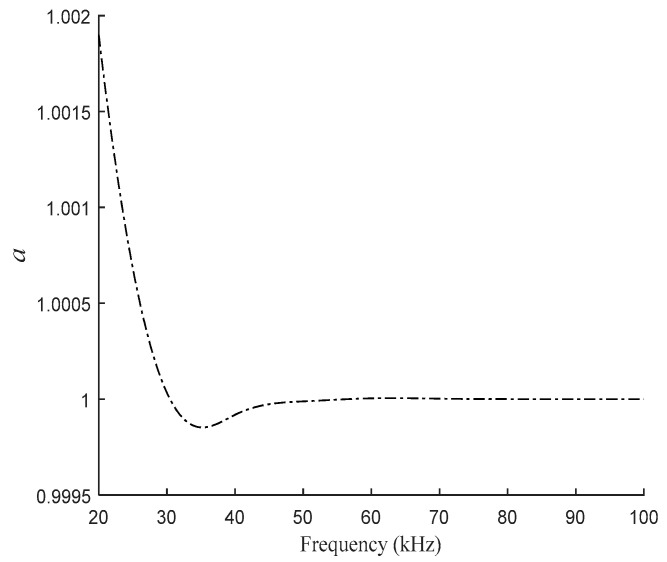
Variation of *a* with frequency.

**Figure 9 sensors-21-06292-f009:**
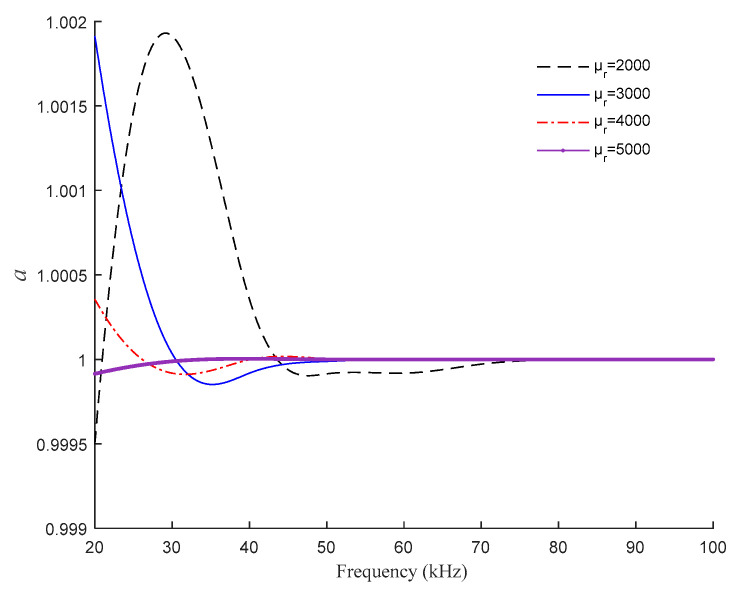
Variation of *a* with frequency under different *μ_r_*.

**Figure 10 sensors-21-06292-f010:**
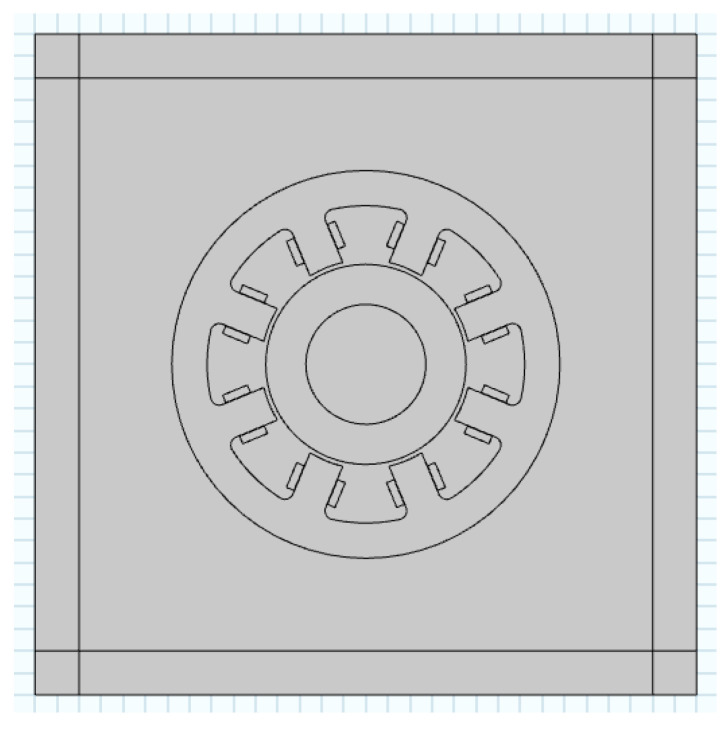
Two-dimensional finite element model of the designed self-inductive displacement sensor.

**Figure 11 sensors-21-06292-f011:**
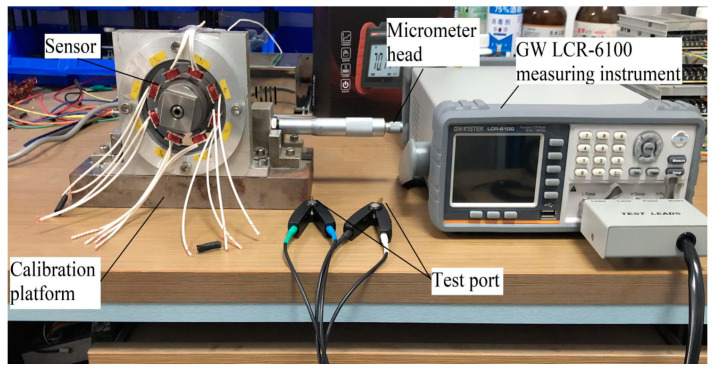
The static measurement experimental device.

**Figure 12 sensors-21-06292-f012:**
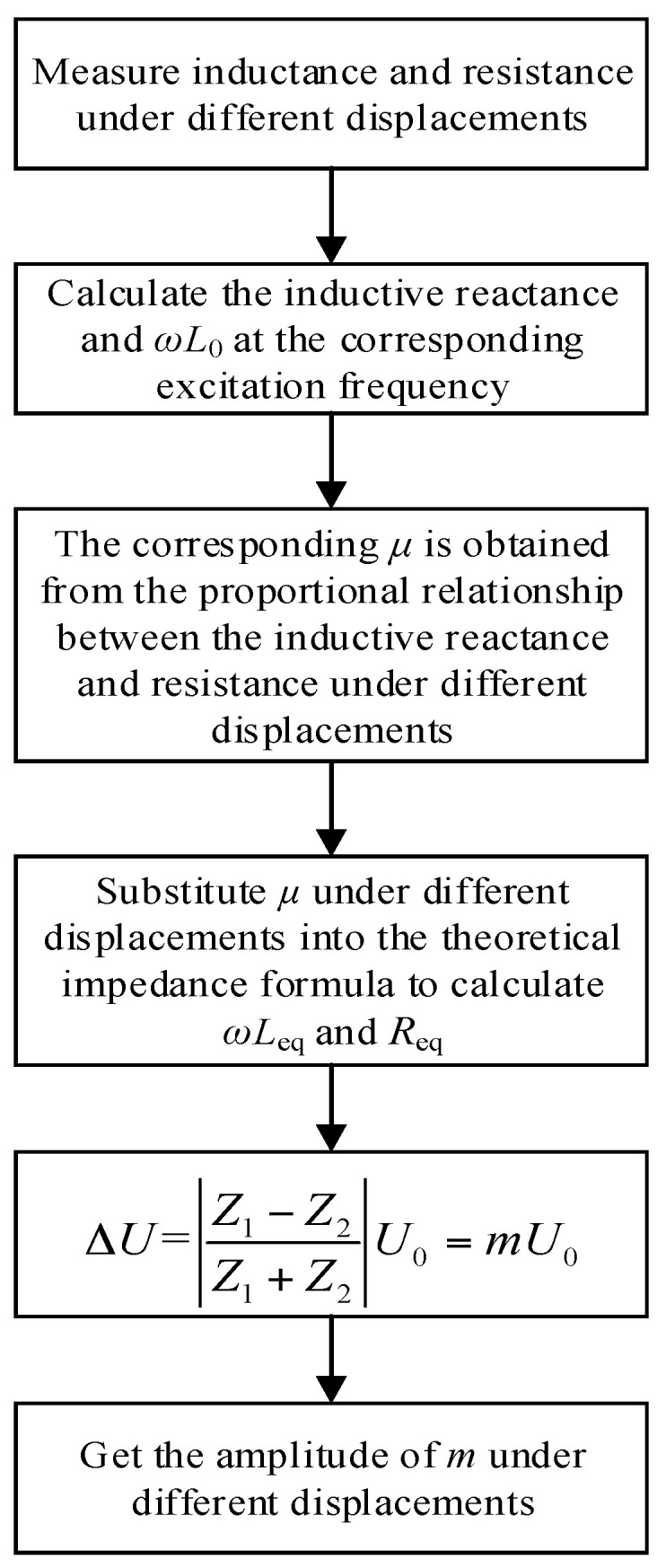
The calculation flow chart of the ratio of output voltage to input voltage under different displacements.

**Figure 13 sensors-21-06292-f013:**
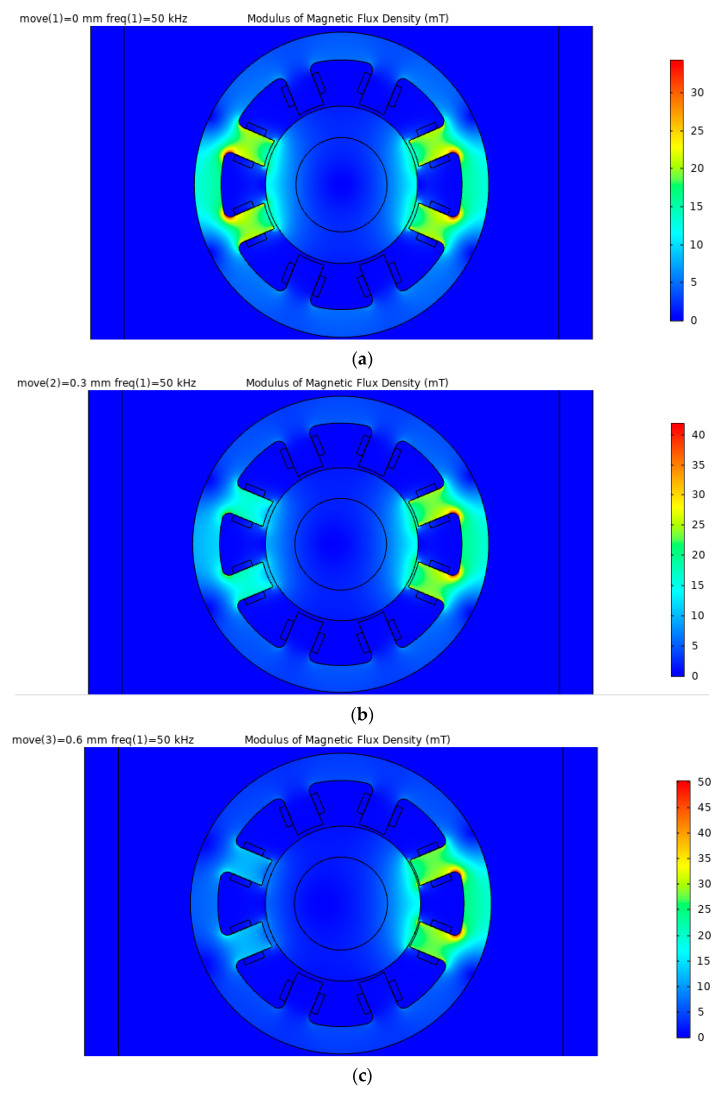
Simulation distribution diagrams of the magnetic flux density modulus of the sensor. (**a**) Δ*δ* = 0 mm; (**b**) Δ*δ* = 0.3 mm; (**c**) Δ*δ* = 0.6 mm.

**Figure 14 sensors-21-06292-f014:**
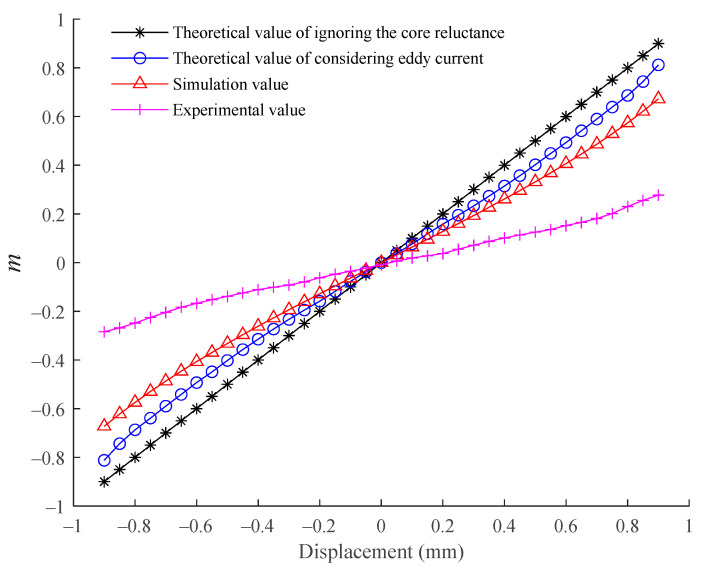
Variation of *m* with the rotor displacement.

**Figure 15 sensors-21-06292-f015:**
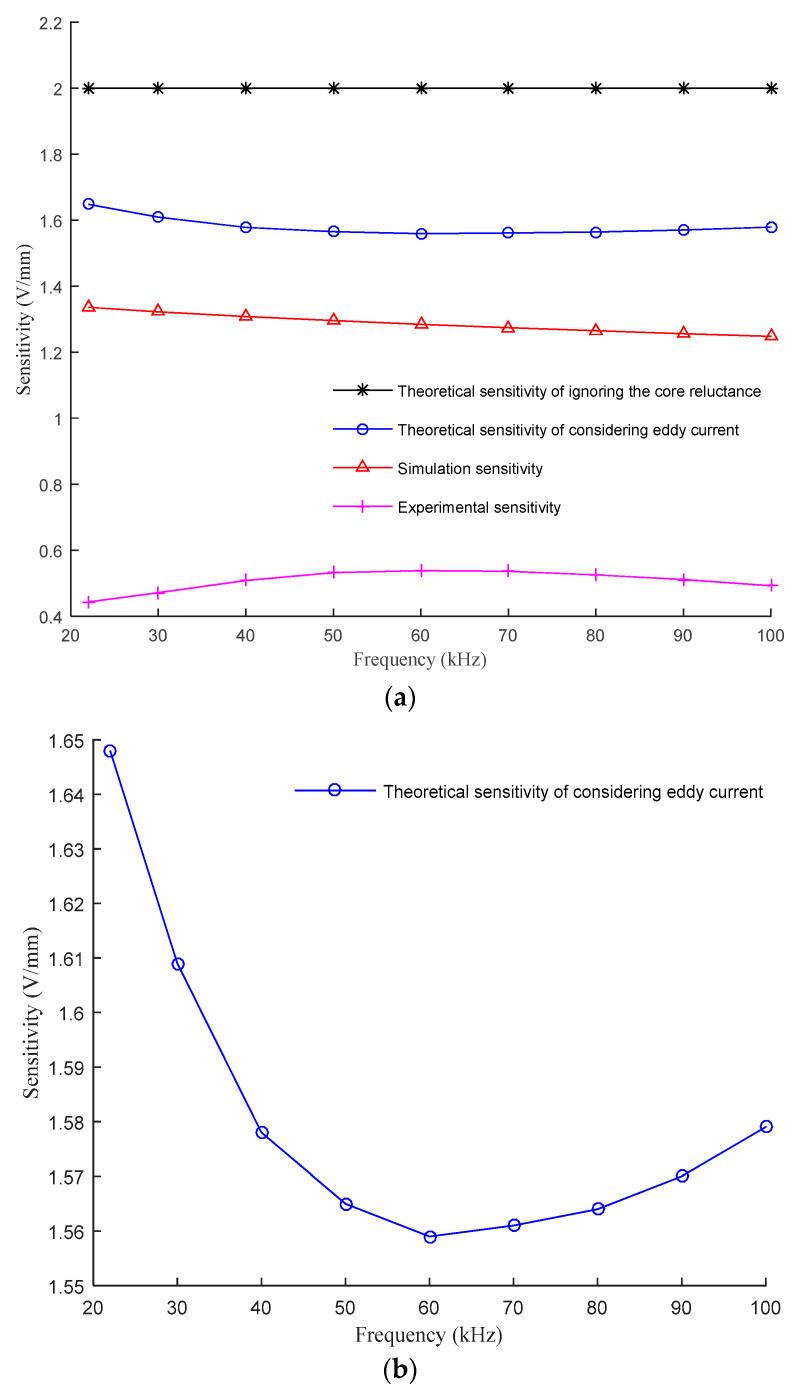
Variation of experimental sensitivity, simulation sensitivity and theoretical sensitivity with frequency. (**a**) Variation of sensitivity with frequency; (**b**) Variation of theoretical sensitivity considering core eddy current with frequency; (**c**) Variation of experimental sensitivity with frequency; (**d**) Variation of simulation sensitivity with frequency.

**Figure 16 sensors-21-06292-f016:**
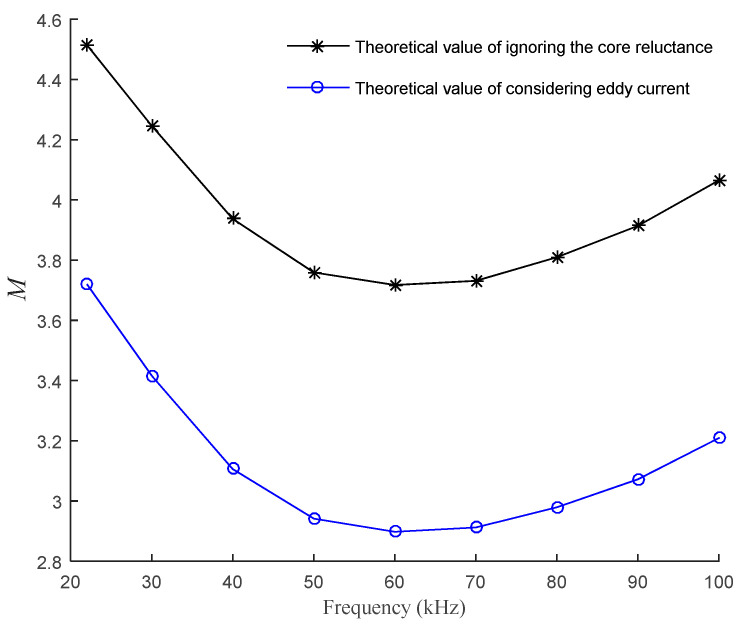
Variation of *M* with frequency.

**Table 1 sensors-21-06292-t001:** Structural parameters of the self-inductive displacement sensor.

*δ*_0_ (mm)	*D* (mm)	*D*_0_ (mm)	*D*_1_ (mm)	*D*_2_ (mm)	*A*_0_ (mm^2^)	*P* (mm)	*N* _1_	*d*_0_ (mm)
1.0	44.2	46.2	72	88	32	8	100	0.25

**Table 2 sensors-21-06292-t002:** Sensitivity and nonlinear error of the experiment.

Frequency (kHz)	Sensitivity (V/mm)	Nonlinear Error (%)
22	0.443	± 2.26
30	0.471	± 2.13
40	0.508	± 2.49
50	0.532	± 2.36
60	0.538	± 2.33
70	0.536	± 2.34
80	0.525	± 2.38
90	0.511	± 2.53
100	0.492	± 3.06

**Table 3 sensors-21-06292-t003:** Sensitivity and nonlinear error of simulation.

Frequency (kHz)	Sensitivity (V/mm)	Nonlinear Error (%)
22	1.336	± 0.37
30	1.322	± 0.37
40	1.308	± 0.38
50	1.296	± 0.38
60	1.284	± 0.39
70	1.274	± 0.39
80	1.265	± 0.39
90	1.256	± 0.40
100	1.248	± 0.40

**Table 4 sensors-21-06292-t004:** Sensitivity and nonlinear error of the theoretical model considering core eddy current.

Frequency (kHz)	Sensitivity (V/mm)	Nonlinear error (%)
22	1.648	± 0.30
30	1.609	± 0.15
40	1.578	± 0.32
50	1.565	± 0.32
60	1.559	± 0.32
70	1.560	± 0.32
80	1.564	± 0.48
90	1.570	± 0.47
100	1.579	± 0.32

## Data Availability

Not applicable.

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
