# Peer review of "Research on Coil Impedance of Self-Inductive Displacement Sensor Considering Core Eddy Current"

_sensors, 2021, doi:10.3390/s21186292_

Round 1
Reviewer 1 Report
The authors have developed a theoretical approach to describe the parameters of a displacement sensor for an active magnetic bearing system. Their approach considers the influence of the magnetic permeability of the core on the sensor output signal. This is its main difference from existing approaches. The article consists of theoretical and experimental parts, its structure is logical. The manuscript is interesting, but not without disadvantages. The reviewer believes some of them are significant.
1) Line 133. μ0 is the permeability of vacuum (or permeability of free space, or magnetic constant), not the permeability of the air, as the authors wrote.
2) Line 133 says “μ1 and μ2 are the relative permeability of stator core and rotor core respectively”. But line 184 says “μ1 is the real part of the relative complex permeability…. μ2 is the imaginary part of the relative complex permeability…”. The authors need to use different designations for these quantities.
3) Not all quantities are described. For example: Z1 and Z2 (Line 219) and μ’* and μ’’* (Line 222).
4) The captions at Fig. 6 are very small.
5) The authors use μr = 3000 without appropriate references.
However, the main disadvantages are as follows.
6) The authors do not mention which values of σ and Ls they used to calculate the dependencies in Figure 6. Moreover, the following questions arise. Do all core laminations have the same size Ls (see Fig. 4)? Do the dependencies in Figure 6 change when Ls and μr change? How does the value of a (Line 269) change in this case?
7) Formula (9) describes the complex magnetic permeability of the core lamination in terms of frequency, size, conductivity and magnetic permeability μr of the lamination material. However, the authors forget that μr is also a complex quantity and depends on the frequency. This can significantly affect the frequency dependencies μ1 and μ2 and the value of a (Line 269). The reviewer believes that it is necessary to consider this or prove that it is irrelevant.
8) According to the authors, μ ’and μ’’ are different (line 282 and beyond). However, according to formula (9), this can only be caused by a change in μr since other values cannot be changed when the rotor is displaced. The change in μr requires a change in the magnetic field through the core. Therefore, it is necessary to discuss in more detail why μ ’and μ’’, according to the authors, are different.
9) As can be seen from Figures 10 and 11, there is a more than threefold difference between model predictions and experimental results. Therefore, the reviewer believes that it is necessary to check all the calculations and make a more carefully analysis of the discrepancies than it is presented in the article.
Consideration of the article can be continued after revision.
Author Response
Dear Dr. Reviewer,
Please see the attachment.
Best regards,
Mr. Zongqiang Ren

Reviewer 2 Report
Authors presented the manuscript which must be improved. The subject is well known but the results are not confirmed by some of the known methods such as FEM, BEM,...
They must give the comparison of their results and those obtained by other methods. They give this comparison bay the tables.
Best Regards,
Author Response

(The authors gave the same response as above.)

Round 2
Reviewer 1 Report
The authors considered all the comments of the reviewer. They proved that the differences between theory and experiment is due to the peculiarities of the experimental setup. The differences are significant, but the article reflects well the spirit of scientific research and has the potential for further development, both theoretical and experimental. The reviewer recommends the article for publication.
Author Response
Dear Dr. Reviewer,
Thank you for your affirmation of our article. We will continue to study in the next work and strive to write higher quality articles.
Best regards,
Mr. Zongqiang Ren
Reviewer 2 Report
Thank you for your correct answer.
Author Response

(The authors gave the same response as above.)
